# B Vitamins and Fatty Acids: What Do They Share with Small Vessel Disease-Related Dementia?

**DOI:** 10.3390/ijms20225797

**Published:** 2019-11-18

**Authors:** Rita Moretti, Costanza Peinkhofer

**Affiliations:** Neurology Clinic, Department of Medical, Surgical and Health Sciences, University of Trieste, 34149 Trieste, Italy; costa.peinkhofer@gmail.com

**Keywords:** small vessel disease, vascular dementia, vitamins B, homocysteine, fatty acids, neuroinflammation, redox

## Abstract

Many studies have been written on vitamin supplementation, fatty acid, and dementia, but results are still under debate, and no definite conclusion has yet been drawn. Nevertheless, a significant amount of lab evidence confirms that vitamins of the B group are tightly related to gene control for endothelium protection, act as antioxidants, play a co-enzymatic role in the most critical biochemical reactions inside the brain, and cooperate with many other elements, such as choline, for the synthesis of polyunsaturated phosphatidylcholine, through S-adenosyl-methionine (SAM) methyl donation. B-vitamins have anti-inflammatory properties and act in protective roles against neurodegenerative mechanisms, for example, through modulation of the glutamate currents and a reduction of the calcium currents. In addition, they also have extraordinary antioxidant properties. However, laboratory data are far from clinical practice. Many studies have tried to apply these results in everyday clinical activity, but results have been discouraging and far from a possible resolution of the associated mysteries, like those represented by Alzheimer’s disease (AD) or small vessel disease dementia. Above all, two significant problems emerge from the research: No consensus exists on general diagnostic criteria—MCI or AD? Which diagnostic criteria should be applied for small vessel disease-related dementia? In addition, no general schema exists for determining a possible correct time of implementation to have effective results. Here we present an up-to-date review of the literature on such topics, shedding some light on the possible interaction of vitamins and phosphatidylcholine, and their role in brain metabolism and catabolism. Further studies should take into account all of these questions, with well-designed and world-homogeneous trials.

## 1. Introduction

Discussion of vitamins and vascular dementia is akin to opening Pandora’s box. Much has been written on vitamin supplementation and dementia, but results are still under debate, and no real conclusion has yet been drawn [1]. Above all, two of the most significant problems have emerged from the debate. The first is that, from the hundreds of relevant studies, no consensus on the application of standard diagnostic criteria has been reached between Alzheimer’s disease (AD) or MCI, nor on the best term to diagnose small vessel disease-related dementia. The second problem is that a possible correct time of implementation has yet to be determined to have the most effective clinical result. Nevertheless, as no specific therapeutic options have been discovered for the two most globally relevant forms of dementia (AD and small vessel disease-related dementia), different risk factors for cognitive impairment have been researched, and vitamin supply and fatty acids could be a potential therapy.

## 2. Vascular Dementia and Small Vessel Disease-Related Dementia

Vascular dementia should be one of the simplest diagnosed pathologies due to the apparent temporal correlation between an acute vascular brain lesion and its onset. Nonetheless, consensus criteria for vascular cognitive impairment remain under debate, which began in 1983 when NINDS-AIREN criteria were written [2]. These criteria have been partially modified by the ICD-0 [3]. Despite multiple attempts, the current clinical diagnostic criteria for vascular dementia are still being debated. They lack a definite morphological substrate for the different types of cognitive disruption due to vascular causes. In fact, three different subtypes have been recognized: Genetic type of vascular dementia (CADASIL or CARASIL), macrovascular type of dementia (multi-infarct dementia or strategic infarct dementia), and microvascular type of dementia (subcortical vascular dementia or, more appropriately, small vessel disease-related dementia) [4,5,6].

The most recent effort to categorize vascular impairment relies on DSM V [7,8]. In the same year, the Standards for Reporting Vascular changes on Neuroimaging (STRIVE) study provided the same guidelines for recommended standards for research on vascular dementia with MRI and CT [9]. For the first time, a panel completed a standard advisory about the terms and definitions for features visible on MRI and minimum standards for image acquisition. Signs of small vessel disease include, in a conventional MRI, recent subcortical infarcts, white matter hyperintensities, lacunes, prominent perivascular spaces, and cerebral microbleeds, with possible consequent atrophy (see below for a more accurate description).

Small vessel disease (SVD) results from damage to the small penetrating arteries and arterioles in the pial and lepto-meningeal circulation, along with penetrating and parenchymal arteries and arterioles, pericytes, capillaries, and venules [10]. The prevalence of SVD increases exponentially with aging. A review of pathologic studies shows enormous differences in the incidence of subcortical vascular dementia (sVaD) ranging from 0.03% to 85.2%, with means around 11%, while in a series of recent autopsies from Japanese geriatric hospitals it was 23.6% to 35%. In Europe, prevalence rates of SVD-related dementia estimated between ages 65–69 to 80+ years ranged from 2.2% to 16.3% [11,12,13,14]. Aging is the most critical risk factor in developing small vessel disease; it leads to the loss of arterial elasticity. This fact causes a reduction of arterial compliance due to the altered mechanical small vessel walls [15]. The effect is the loss of the autoregulatory capabilities of cerebral arteries; as a consequence, the brain suffers from a higher vulnerability to hypotension, with possible major ischemia, when the systemic blood pressure dips below a critical threshold [16,17,18]. Arteriosclerosis and cerebral amyloid angiopathy (CAA) [19,20,21,22] are the most crucial pathogenic expressions of small vessel disease. The mechanism here described is supported by the concomitant age-related impairment of vascular autoregulation; it has been described as the low-level functioning of the autonomic nervous system, with direct and endothelium-mediated altered baroreflex activity [23,24,25,26]. Small vessel disease, due to the primary localization of the white matter hyperintensities, may affect the integrity of the adjacent medial cholinergic pathway [27] or could develop from the deafferentation of the basal forebrain cholinergic system to the tubero-mamillary tracts [28,29], observing, therefore, perverse procrastination of the hypoperfusion. Neural activity is typically supplemented by increased blood flow, originated by a retrograde vascular dilatation of upstream arterioles outside the activated area [30]. The altered “retrograde vasodilatation system” (which occurs in small vessel disease-related dementia) could be another factor that worsens vessel dysregulation [30].

Cerebral small vessel disease includes a neuroimaging and a combined pathological description, which comprise different imaging changes in the white matter and subcortical grey matter, including small subcortical infarct, lacunes, white matter hyperintensities (WMHs), prominent perivascular spaces (PVS), cerebral microbleeds (CMBs), and atrophy. Lacunes derive from a complete occlusion of the small arteries, leading to an infarct in a specific area (basal ganglia, capsule, pons). Moreover, SVD is characterized by an associated hypoperfusion progression, causing incomplete ischemia of the deep white matter [31,32,33,34] accompanied by inflammation, diffuse rarefaction of myelin sheaths, axonal disruption, and astrocyte gliosis [26]. In small vessel disease, occlusion of the deep periventricular-draining veins is also evident [35], with the disruption of the blood-brain barrier (BBB) as a consequence. The BBB disruption leads to a consequent leakage of fluid and plasma cells, which eventually potentiates the perivascular inflammation, the demyelination process, and gliosis, gathering a multifactorial genesis for white matter alterations [36,37,38]. Cerebral small vessel disease is what we define to as a “progressive disease” [26]. Lesions progress over time, and the long-term outcome and impact on brain damage vary. Recent studies indicated that the strongest predictor of white matter progression is the high density, at baseline, of the white matter hyperintensity, with a rapid confluence of the lesions [39,40,41,42]. Nonetheless, one of the most unsolved doubts is which is the determinant factor for an acceleration process that mediates the transition from small vessel disease toward the subcortical dementia process? The burden of lacunes and a profound amount of white matter alterations (WMLs) [43,44,45] are two suitable candidates.

The cognitive alterations are determined by the specific point-to-point impairment, but also by the interruptions of the frontal and prefrontal-thalamus and thalamus-frontal and basal forebrain networks [46,47], which can also lead to functional cortical deafferentation. The caudate nucleus is the most precociously affected region by chronic hypoperfusion, followed by the putamen, insula, precentral gyrus, inferior frontal gyrus, and middle frontal gyrus. All these regions require, at steady state, more than 20% of the metabolic request compared to other brain areas [48,49,50,51,52,53,54,55]. Reduced metabolic rate of oxygen (estimated of about 35% in white matter) [56,57] has been found in patients with small vessel disease-related dementia; moreover, an incongruity between the brain oxygen supply and its consumption has been described in sVaD, which determines an altered neurovascular coupling and altered vasomotor reactivity [26,58,59,60,61,62,63].

Neuropsychological pattern profiles of sVaD are related to the subcortical-cortical loop deafferentation and are distinguished by poor executive function, poor planning, working memory alterations, loss of inhibition, reduced mental flexibility, multitasking procedures invalidation, and decrease speed of executive process [64,65,66,67,68,69,70].

No specific treatment has been discovered, either as pathogenic or highly standard recommended for this condition.

## 3. State-Of-The-Art: What Do We Know about Nutrients and Brain Degenerative Diseases?

Decades have passed since the first descriptions of how vitamins can affect brain networks [71], but results remain compelling. Cummings [72] suggested that the nutraceutical approach might have an effect on synaptic integrity, and therefore, it could be useful in many different clinical conditions, such as AD, vascular disease, traumatic brain injury, and Parkinson’s disease. Choline is an essential nutrient for humans. It is a precursor of the synthesis of all the membrane phospholipids (e.g., phosphatidylcholine (PC)), the neurotransmitter acetylcholine, and via betaine, the methyl group donor S-adenosylmethionine. High choline intake during gestation and in the soon-after postnatal period, in animal models, improves cognitive function in adulthood. It has been associated with a prevention of age-related memory decline, and of the neuropathological changes associated with AD, epilepsy, fetal alcohol syndrome, and inherited conditions such as Down and Rett syndromes, but also with stroke and, recently, vascular-risk conditions [73,74]. Choline and its metabolites are differently associated with cardiometabolic risk factors, history of cardiovascular disease, and MRI-documented cerebrovascular disease in older adults. [75]. Choline might modify the DNA methylation in the brain and may induce alterations in the expression of genes that encode proteins essential for learning and memory processing, suggesting a possible epigenomic mechanism of action [75]. Prenatal choline supplementation has been demonstrated to show a three-day advancement in hippocampal development [76]. This effect on adult neurogenesis has been observed together with a local increment of other trophic factors: nerve growth factor, brain-derived neurotrophic factor, insulin-like growth factor 2, and vascular endothelial growth factor [77,78]. A concomitant enlarged dimension of the basal forebrain cholinergic neurons has also been observed [79], with a concomitant increase of acetylcholine production [80].

Choline influences, as a cofactor of methyl donors, the DNA methylation process (in addition to folate and vitamin B12; see elsewhere in this manuscript). Choline is a strong determining factor of transcriptional regulation and the methylation process of different regions of insulin growth factor 2 gene [81,82], and many others fundamental for synaptic plasticity, with the cascade of dynamic processes highly dependent on it, i.e., learning and memory [83,84,85,86,87,88].

Choline also is a precursor of phosphatidylcholine, the principal constituent of all biological membranes, including the polarized ones, such as neurons and astrocytes. Evidence that phospholipid metabolism is abnormal in AD originated with postmortem brain sample studies [75,89]. These studies showed reduced levels of phosphatidylcholine and phosphatidylethanolamine, and increased levels of their metabolites, glycerophosphocholine, and glycerophosphoethanolamine, in the cerebral cortex of AD patients compared to age-matched controls [89,90]. These variations have also been observed in brain regions free of plaques and tangles [90]. Therefore, this data suggests that the defect does not rely on an amyloid-related process but is widespread through the AD brain. More recently, reduced levels of phosphatidylcholine and phosphatidylethanolamine and increased levels of their metabolites, glycerophosphocholine and glycerophosphoethanolamine, have been reported in vascular-risk factors conditions, such as fatty liver disease [91], obesity, and insulin resistance [92,93].

Furthermore, a substantial reduction in molecular species of phosphatidylcholine containing docosahexaenoic acid (PC-DHA) levels has been observed in the temporal cortex gray matter of AD patients [94]. These markers have been observed in AD brains but also AD peripheral blood plasma [95,96,97]. All these findings support another theory for AD pathogenesis: Lipid altered or defective production [75,98,99]. Healthy old subjects with a lower level of erythrocyte phospholipid n-3 fatty acid (eicosapentaenoic (EPA, 20:5n-3)) are more prone to faster cognitive alteration, compared to healthy old subjects with normal levels of erythrocyte phospholipid n-3 fatty acid (eicosapentaenoic (EPA, 20:5n-3)) [100]. Docosahexaenoic acid and erythrocyte phospholipid n-3 fatty acid (eicosapentaenoic acid) are transported through the BBE as lysophosphatidylcholines by a specific transporter [101,102]. Therefore, it has been argued that low plasma levels of lysophosphatidylcholine might reduce fatty acids inside the brain in AD patients [97,103].

Hence, medical supplementation of food, including with phosphatide precursors (docosahexaenoic and eicosapentaenoic acid, phospholipids, choline, uridine monophosphate, vitamin E, vitamin C, and vitamins B12, B6, and B9) has been employed in different clinical trials [104,105]. The second study has two outcomes: The primary aim is to demonstrate an improvement of memory scores in specific tests in mild AD when compared to placebo. Secondary outcomes include different sub-scores of neuropsychological battery and EEG measures, including relative and absolute power in alpha, beta, theta, and delta frequency bands, associated with the phase lag index in each of the frequency bands. The study achieved its primary outcome, showing significantly better performance in memory in the treated mild AD group. It did not obtain any significant measure in the second neuropsychological outcome, but showed that peak frequency and phase lag index of the delta frequency showed a drug/placebo difference in favor of the treated group [105]. Three other studies continued from these finding [106,107,108]. Rijpma et al. [106] demonstrated that prolonged diet supplementation could increase circulating levels of fatty acid levels and uridine monophosphate. Uridine, together with docosahexaenoic and eicosapentaenoic acids, is a rate-limiting precursor via the Kennedy pathway for the synthesis of phospholipids in neuronal membranes.

A recent review of the studies on food supplementation [109] has drawn hopeful conclusions: early interventions work better, and mildly affected patients improve memory scores in specific tests. Moreover, their caregivers perceive an amelioration (even if not significantly) of AD behavior, and these supplementations might potentially delay or ameliorate the pathogenetic process in AD and small vessel disease [110,111,112,113].

## 4. Vitamin B1 (Thiamine)

Thiamine is probably one of the most studied vitamins related to Wernicke encephalopathy and Korsakoff syndrome, which have been thoroughly studied for decades [114,115]. Nevertheless, recent data emphasizes the role of thiamine, due to the massive increase of alcoholism in Western countries [116,117,118], in addition to the finding that thiamine defects are frequently found in association with dialyzed patients, hyperemesis gravidarum, malignancy with or without therapy, sleeve gastrectomy, magnesium depletion, AIDS, etc. [119,120,121,122,123].

Thiamine, also called vitamin B1, is a labile quaternary ammonia compound, not produced in humans. It is found as unphosphorylated thiamine (i.e., free TH) and as phosphorylated derivates, with magnesium as a cofactor: Thiamine monophosphate (THMP), thiamine diphosphate (THDP, aka TH pyrophosphate), and thiamine triphosphate (THTP) [124,125].

Thiamine acts as a cofactor in three key-enzymatic reactions: The conversion of pyruvate to acetyl-CoA, the conversion of alpha-ketoglutarate to succinate, in the Kreb’s cycle, and the catalysis by transketolase in the pentose monophosphate shunt [125]. Lack of alpha-ketoglutarate dehydrogenase interrupts the Kreb’s cycle, essential for ATP production in the brain; the loss of ATP induces a quick increment of the calcium inflow, with consequent induction of neuronal apoptosis and an abrupt growth of the glutamate currents [125,126]. The loss of alpha-ketoglutarate dehydrogenase causes a decrease of the aspartate and GABA brain concentration, a decrement of the reduced oxidative phosphorylation, and an increment of lactate [127,128,129,130]. Diminished transketolase activity leads to a loss of sphingolipid synthesis, an overproduction of branched-chain amino acids, and an altered pentose phosphate shunt [131,132,133,134]. Until the last decade, there were limited works describing the causative role of brain damage induced by thiamine loss per se; usually, thiamine defects have been tightly related to alcohol addiction, the former being recognized as the causative factor of the neural alterations, rather than the alcohol by itself.

Moreover, it has been observed that in alcohol-addiction there is an altered gene expression of two proteins, carriers for thiamine, THTR1 and THTR2 (not known if genetically or epigenetically transformed) [135,136,137], suggesting a possible superimposed damage factor. Thus, genetic predisposition of thiamine transporter reduced efficacy is currently more accepted as a potentiating effect of brain damage, even in well-nourished patients. Animal models have been used to prove that severe isolated thiamine deprivation brings a significant alteration of the inferior colliculus and the medial vestibular nucleus over a one-week period. More prolonged thiamine depletion caused severe basal ganglia impairment and, in the final period of observation, functional alteration of the mammillary bodies and the dorsal medial nucleus of the thalamus [138]. Some studies suggested a possible relationship between a loss of thiamine and a macroscopic decrease of the endorphinergic system [139], but this data remains isolated because it has been observed in many patients, with a contemporary loss of thiamine and heavy alcohol consumption. The principal cause of the observed data remains a mystery (alcohol induction of endorphin depletion has been positively documented) [130]. The loss of the activity of the three thiamine enzymes in the brain (the conversion of pyruvate to acetyl-CoA, the active participation in the conversion of alpha-ketoglutarate to succinate, in the Kreb’s cycle, and the catalysis by transketolase in the pentose monophosphate shunt) has been observed in many neurodegenerative conditions [140,141,142,143,144]. The administration of benfotiamine induced a mild inhibition of cholinesterase, leading to a significant amount of acetylcholine in the post-synaptic space and determining a reduction of amyloid plaques and tau tangles, probably reducing the cascade of neuroinflammation, which promotes the hyperphosphorylation process [145,146,147,148].

Thiamine also has a non-coenzymatic function [149]; its effects on the axonal conductance and the release of different neurotransmitters, such as acetylcholine, dopamine, and noradrenaline, has been described [149,150,151]. These data have been supported by the findings of a rapid change of thiamine status (the so-called mobile thiamine pool) [150,151] in different neural networks. In particular, oxythiamine stimulates potassium-evoked acetylcholine release in the presence of calcium [152,153]. By definition, thiamine exerts a refining system able to protect the hippocampal neurons cultured with an excess of glutamate [154]. On the contrary, thiamine deficiency also induces an excess of glutamate release, diminished by a glutamatergic blocking action with an N-methyl-D-aspartate (NMDA) antagonist (similar to memantine) [155,156]. These studies have not yet been applied in clinical practice [157].

## 5. Vitamin B2 (Riboflavin)

Riboflavin, or B2, is mainly dependent on dietary intake. It is active in different forms, as riboflavin, or as coenzymes for several reactions, namely, flavin mononucleotide (FMN) and flavin adenine dinucleotide (FAD). Riboflavin is involved in a large variety of processes and has a critical neuroprotective function [158]. It is a primary antioxidant; thus, B2 is involved in glutathione reduction, where FAD is a co-factor, forming reduced, active glutathione, which acts against oxidative stress and lipid peroxidation. Vitamin B2 increases the activity of antioxidant enzymes, such as the superoxide dismutase (SOD) and catalase [158,159,160,161], additionally. Riboflavin seems to also reduce the oxidative damage after reperfusion, through a direct action on free radicals, as shown by studies on rabbits’ hearts after infarction [162], mice ischemic liver cells [163], and rats’ brains [164]. In these cases, vitamin B2 reduces edema and neuronal death after traumatic brain injury [151,160,164,165]. Riboflavin, together with folate, lowers homocysteine levels, hence helping to avoid vascular and toxic damages: FAD is needed in the one-carbon metabolism cycle to reduce 5,10-methylenetetrahydrofolate (5,10-MTHF) to 5-methyl THF which, in turn, provides the methyl group for homocysteine re-methylation to methionine [158,161,166]. This vitamin might play a role in the reduction of glutamate excitotoxicity via direct inhibition of glutamate neuronal release, as evidenced in animal studies [167,168]; further, it is crucial for the tryptophan-kynurenine pathway, where neuroactive compounds that might influence glutamate receptors are produced [158]. In addition, B2 seems to have a direct anti-inflammatory ability, through inhibition of NF-kB and high-mobility group protein B1 (HMGB1) [158,169], and it is involved in mitochondrial functioning, FAD and FMN being co-factors for complex I and complex II of the electron transport chain [158,159]. Lastly, some of these functions are shared with the B6 vitamin; B6 requires the active form of B2, FMN, as a co-factor, in order to be transformed into its active form pyridoxal 5′ phosphate (PLP) [170].

Riboflavin deficiency derives from inadequate dietary intake, mainly derived from a rice-based diet, poor animal proteins [171], or in conditions of higher demand, e.g., pregnancy, childhood, aging [159], genetic disorders, or malabsorption conditions. Deficiency is also related to drug and substance interaction, e.g., furosemide or alcohol, contributes to riboflavin reduction [172]. Lack of riboflavin is recognized in starvation and lower income groups, but it has also been found in adolescents with low milk consumption [166], and a recent study showed that in the Western world, including Europe, North America, Australia, and New Zealand, 40% of people older than 65 years are below the estimated average requirement [173]. A suboptimal, subclinical level of B2 is likely more common than previously thought and might affect all ages [174]. Further studies, based on biomarkers more than simple dietary intake surveys, are required for more reliable data and a better knowledge of this crucial vitamin deficiency [170].

Ariboflavinosis may lead to several conditions, such as anemia, cheilosis, glossitis, angular stomatitis, cataract, and seborrheic dermatitis [159,174,175]. It is frequently associated with other vitamin deficiencies, and signs and symptoms are rarely isolated [170]. Impaired nerve function has also been correlated to riboflavin insufficiency, both as acquired [174,176,177] and genetic disorders, i.e., Brown–Vialetto–Van Laere neuropathy [178].

Subclinical levels of riboflavin might be related to different neurological diseases, such as Parkinson’s disease, migraines, and multiple sclerosis, due to the multiple roles discussed above [158,161]. Thus, when implemented, riboflavin seems to have neuroprotective functions, and clinical trials have reported how a high intake of this vitamin reduced motor impairment in patients with Parkinson’s disease (PD) and proved to be useful for migraine prophylaxis in adults [179,180,181].

On the contrary, the direct relationship between riboflavin and dementia, mainly the so-called small vessel disease/subcortical vascular dementia, has been rarely investigated, even if vitamin B2 is related to endothelial release of nitric oxide [166] and to the catabolism of homocysteine, both of which are related to cerebral small vessel disease [159,182,183,184,185].

Some studies have analyzed the relationship between B2 and cognitive decline [186,187,188,189] without investigating possible direct molecular mechanisms. In one study, the authors enrolled a cohort of 70-year-old subjects, who had previously been assessed for intelligent quotient at age 11 years. At 70 years, participants underwent a range of neuropsychological tests, including the Mini-Mental State Examination (MMSE), and the National Adult Reading Test (NART), and completed food questionnaires regarding the intake of several nutrients, including B vitamins and riboflavin [187]. No significant association was found between diet and cognitive performances [187]. In line with this, a Korean observational study [188] explored the association between intake of B vitamins, including riboflavin, and cognition in three groups of subjects over 60 years: Normal, mild cognitive impairment, and with Alzheimer’s disease. Riboflavin intake was positively correlated to cognitive test scores, e.g., MMSE for Koreans (MMSE-K) and the Boston Naming Test, in both the AD group and the MCI group, whereas no correlation was found in healthy subjects [188]. Of note, there was no significant difference in vitamin intake among the three groups when comparing AD patients with controls, as also noted by other authors [190]. Another Korean study found an association between poor cognitive performance, scored with MMSE-K, and riboflavin intake [189], although herein, the correlation was also positive in subjects with average scores. Two earlier studies [191,192] investigating subjects free of cognitive impairment reported a better memory function with higher riboflavin levels and an absence of correlation between cognitive performances and B2.

These results show how riboflavin might be related to cognition, although the reasons are yet to be proven. Two well-established risk factors for cerebrovascular damage and small vessel disease are hyperhomocysteinemia, acting via endothelial impairment, ischemia, and oxidative damage [182], and advanced glycation end product (AGE) formation, which can lead to micro and macrovascular damage, accelerating AD [193]. Riboflavin is an independent determinant of homocysteine levels [184,194], and its deficiency is related to higher homocysteine levels, increasing vascular complications. B2 is also essential for vitamin B6 activation. The latter is a powerful anti-glycation agent, preventing AGE formation, thus protecting blood vessels [195,196,197]. Furthermore, these findings suggest that B2 levels might differ even with a similar dietary intake [166,179]. However, additional studies, using more robust biomarkers, i.e., plasma levels instead of food intake questionnaires, and experimental approaches are required to better evaluate the role of B2 in cognitive decline.

## 6. Vitamin B3 (Niacin)

The vitamin B3 family consists of three different molecules: Niacin or nicotinic acid, nicotinamide, and nicotinamide riboside (NR) [198,199]. Together with tryptophan, they are essential for the synthesis of nicotinamide adenine dinucleotide (NAD+) and nicotinamide adenine dinucleotide phosphate (NADP+), and their reduced forms NADH and NADPH [198]. These co-factors play a fundamental role in several cellular processes and are essential for cells’ survival [198,199,200]. They are involved in vital redox reactions, such as glycolysis and gluconeogenesis, beta-fatty oxidation, and steroid anabolism, thus, protecting cells from oxidative damage. They are crucial for mitochondrial respiration and ATP formation, and are strongly demanded in the citric acid cycle. The vitamin B3 family interacts in the inflammatory cascade, promoting calcium signaling and acting as a direct neurotransmitter via the purine receptor.

Furthermore, the NAD+ pool influences genetic stability and epigenetic variability; it is the main co-factor of poly (ADP-ribose) polymerases (PARPs), ADP-ribose transferases (ATRDs), and sirtuins. These enzymes regulate the DNA repair process, transcription, chromatin expression, cellular death, and senescence regulating some vias, e.g., the telomere length [198,199,200,201].

Vitamin B3 compounds derive from different food sources, such as beans, meat, fish, milk, mushrooms, and enriched flour. However, malnutrition, alcohol abuse, and infectious/autoimmune processes might cause insufficient vitamin levels. The main consequence of acute B3 deficiency, leading to a lack of NAD, is pellagra. This disease is mostly present in emerging nations, where starvation or very poor corn-based diets are dominant. In countries with niacin-fortified food (i.e., flour and grain), pellagra can be only observed in alcohol addicted, immunosuppressed, and cachectic patients. Symptoms of pellagra include diarrhea, dermatitis, and neurological disorders such as anxiety, depression, and dementia [199,201,202].

Due to the above-explained different functions exerted by vitamin B3, suboptimal B3 levels might lead to cellular dysfunction, including early cell death, changes in cellular metabolism, and aging, increasing the chances of misfolding and telomeric alteration, even promoting neoplastic degeneration [199,203]. Some subgroups, such as the elderly, pregnant women, or patients undergoing DNA damaging treatments, are at a higher risk of suboptimal levels because of increased body requests, despite an adequate vitamin intake [199].

The role of B3 in the brain has been studied for more than 70 years [204,205,206], but only recently has the focus shifted from considering vitamin B3 as a determinant factor for cell survival, to the extension to be an essential neuronal and vessel protector [200,201,202,203,204,205,206,207,208,209]. Some of the most critical neuroprotective mechanisms have been primarily investigated in animal models but not yet reproduced in humans. It has been shown that B3, mainly as NAD+, might reduce the lesion size in global [210,211] or focal cerebral ischemia [199,200], as well as in transient cerebral ischemia [212,213], and might ameliorate the primary outcome, after traumatic brain injury [214,215,216,217,218]. Indeed, neuronal death, derived in acute conditions by necrosis and apoptosis, could be prevented via several mechanisms, such as oxidative stress protection [219,220,221], control over PARP-1 activity thus preserving energy levels [211,222], and direct inhibition of pro-inflammatory cytokines [223]. Further, B3, primarily as NR, protects neurons from axonal degeneration, derived from excitotoxicity, due to significant ischemic condition [224].

The same neuroprotective neuro-vascular pathways seem to also influence neurodegenerative pathologies [225]. Animal models show niacin’s role in Huntington’s disease (HD) [226], Parkinson’s disease [227], and Alzheimer’s disease [228]. It is likely that in PD and HD, nicotinamide supply might increase NAD+ and ATP levels, improving mitochondrial functioning, impaired in both pathologies [229]. Additionally, PD patients benefit from B3 supplementation, which reflects the improvement of some symptoms due to its interaction with the NIACR1 receptor, enhancing the availability of NAD+ by either the use of a diet supplemented with NAD+ precursors or the inhibition of NAD+-dependent enzymes, such as PARPs, which compete with mitochondria for NAD+. This could be a viable approach to preventing neurotoxicity associated with mitochondrial defects [201,230,231,232].

In dementias, the neuroprotective action of niacin is related to its effects on cerebral microvascular endothelial cells (ECs) and neurons. It protects both types of cells from oxidative damage, ischemic insults, and senescence, and influences inflammatory processes, primarily related to microglial functions. ECs are basal cells to brain homeostasis and regulate both blood flow and the blood–brain barrier [200,233]. They are injured in clinical stroke but are also altered in subclinical, chronic ischemic insults. Experimental studies on rats have shown how ischemic/hypoxic or direct damage from nitric oxide (NO) induces apoptosis in ECs [233,234]. Niacin protects ECs and neurons from apoptotic injury, preventing the exposure of membrane phosphatidylserine (PS) residues, which act as a phagocytosis signal, inducing the DNA fragmentation [200,207]. B3 interacts with different pathways that activate apoptosis, including caspase 1, 3, 8, PARP-1 activity and protein kinase B (Akt). The latter maintains mitochondrial polarization, avoiding cytochrome C release, and inhibiting the activation of the caspase [207,234].

More interestingly, exposure of PS leads to a loss of anticoagulant membrane components, increasing the risks of secondary thrombosis, clot aggregation, and inflammation [200,207,229]. Furthermore, it has been demonstrated that niacin also reduces atheroma formation, decreasing the total amount of lipoproteins and increasing the incidence of high-density lipoproteins (HDL) [235].

Dysfunctional ECs are determined more often by the normal processes of senescence. Indeed, dysfunction leads to an altered proliferation of ECs, associated with an increment of pro-inflammatory factors, impaired proliferation, higher rates of apoptosis, and increased permeability of the BBB due to the loss of the tight junctions [236,237,238]. Sirtuin1, a component of the sirtuins family, is an NAD+ enzyme involved in several functions, such as the senescence process, gene expression/regulation, and cell death, and it is primarily represented in neurons and ECs. A single study showed a decline in Sirt1 expression and activity in aged mice and aged humans’ ECs in vitro [236]. In this study, the authors demonstrated the underlying relationship between Sirt1 depletion and BBB increased permeability [236]. Since Sirt1 depends on NAD+ activity, one animal study showed that nicotinamide supplementation reversed age-related vascular EC dysfunction [239]. In humans, a phase II randomized clinical trial investigated the effects of Resveratrol, a Sirt1 agonist, in AD patients with promising results [240], but more studies are needed [241].

To the best of our knowledge, we might hypothesize that low B3 levels are frequent in the elderly, and this may influence the development of small vessel disease and subcortical vascular dementia, switching ECs towards a procoagulant state, increasing plaque deposition, reducing Sirt1 expression, and promoting senescence mechanisms, such as apoptosis, neuroinflammation (through Sirt1 and PARP, 200, 242, 243), ECs, and neuronal death. Niacin modulates neuroinflammation in several ways. It regulates microglial activation, known to promote oxidative stress and inflammation, avoiding PS exposure, a trigger of microglial function and proliferation [200]. It modulates PARP-1 activity, involved in NF-kb transcription, which, in turn, regulates the secretion of cytokines and chemokines, and it may also inhibit some pro-inflammatory cytokines, such as interleukin (IL)-1β, IL 6, IL8, and TNF-α directly [223,242].

Nicotinamide restriction has been shown to increase NADPH oxidase, and reactive oxygen species (ROS) release in human keratinocytes [243]. High ROS levels are associated with the oxidative cascade of events, determining many neurodegenerative disorders, due to neuronal and EC loss [244,245]. Nicotinamide has proved to be a powerful antioxidant in animal models, maintaining brain mitochondrial homeostasis and reducing neuronal and EC oxidative stress [220,239,242].

Nicotinamide supplementation has been tested in animals and humans with AD yielding promising but still incomplete results. In AD model mice, NR prevented Aβ accumulation and hippocampal astrocyte loss [228,246,247], attenuating cognitive decline and improving selective cognitive impairment [228,246]. The same benefits have not been confirmed in clinical trials yet, despite past and ongoing attempts [242]. Further, dietary intake of B3 was associated with lower cognitive decline and seemed to have protective effects on the development of AD [248]. More clinical trials are currently investigating nicotinamide supplementation in AD patients and will soon add new knowledge regarding its efficacy in neurodegenerative diseases. However, since vitamin B3 is involved in vital pathways of both neurons and cerebrovascular endothelial cells, an association with dementia is likely, and its supplementation in subclinical conditions might improve not only AD but also vascular dementia outcomes and disease progression [249].

## 7. Vitamin B5 (Pantothenic Acid)

Vitamin B5, also called pantothenic acid/pantothenate, is the primary precursor of coenzyme A (CoA), crucial in several cellular processes. As wither the direct form (CoA) or the acetylated form (acetyl-CoA), this coenzyme is involved in energy production and respiration, via the citric acid cycle. It is essential for fatty acid synthesis and β-oxidation, and for cholesterol, lipid, and sphingolipid biosynthesis, as well as for the production of steroid hormones and neurotransmitters, i.e., acetylcholine [195,201,250,251]. Further, acetyl-CoA plays a role in global histone acetylation, modulating gene expression, cell growth, and proliferation [252,253]. Pantothenic acid, due to the increase of CoA, exerts an antioxidative property [254,255,256] and influences inflammatory factors, such as C-reactive protein (CRP) [257]. Indeed, it has been shown in vitro that pantothenic acid might enhance glutathione levels [254,255,256] and, in animal models, high concentrations protect neurons from radiation damages [255].

Moreover, pantothenic acid exerts an antioxidative pathway through cysteamine, a product of CoA degradation. This catabolic reaction is handled by a family of enzymes, called vanins, and vanin-1 is upregulated in oxidative stress. Hence, it seems plausible that the presence of high CoA turnover in some tissues demonstrates a protective role against oxidative injury [253].

Some pantothenic acid derivatives, such as panthenol and pantetheine, the dimeric form of endogenous pantetheine produced from pantothenic acid and cysteamine, modulate and interact with the cholesterol metabolism, lowering low-density lipoproteins (LDL) and enhancing HDL [258,259,260]. The exact mechanism underlying these processes is still unknown, and a correlation with homocysteine has been hypothesized, involving both CoA biosynthesis and atherosclerosis [259].

Vitamin B5 is present in almost every food, being an essential compound for all forms of life [253,261]. Significant sources of pantothenic acid are broccoli, meat, whole grains, cereals, and avocado. It is worth mentioning that commensal bacteria, such as *Escherichia coli*, can produce and secrete pantothenate directly in the gut, increasing the availability of this vitamin [195,253,255]. Therefore, despite low oral intake, B5 deficiencies have been rarely described in humans. The described symptoms of pantothenic acid deficiency include numbness and paresthesia, insomnia, and irritability, as well as nausea, vomiting, and non-specific cardiovascular alterations [195,201,261].

Pantothenic acid is well represented in the central nervous system (CNS) and is more concentrated in the brain than in the plasma [255,262]. Several old studies focused on its transportation and function in the CNS [263,264,265,266,267], but it has not been investigated as much as the other B vitamins, likely because very few studies described its deprivation.

Vitamin B5 is known to be related to a rare neurodegenerative autosomal recessive disorder, called pantothenate kinase-associated neurodegeneration (PKAN), due to genetic mutations in the pantothenate kinase-2 (PANK-2), an enzyme responsible for the phosphorylation of pantothenate, the first reaction of the CoA biosynthetic pathway [250,268,269,270,271].

Nevertheless, a recent study demonstrated that low levels of B5 were found in post-mortem brain tissue of Huntington’s disease (HD) patients as opposed to matched healthy controls [252]. The authors hypothesized that a B5 deficiency might be involved in the alteration of citric acid cycles, leading to inadequate energy levels, as frequently reported in HD patients.

Additionally, investigators have speculated on the role of acetyl-CoA. Acetyl-CoA is fundamental to several cellular pathways, including gene expression, which is strictly dependent on CoA levels, and its activity might be impaired due to a B5 insufficiency [252].

Only one study has tried to assess the role of pantothenic acid in cognition with unexpected results [272]. Indeed, the authors found that a higher dietary intake of B5 correlates to an increased burden of cerebral amyloid in patients with subjective cognitive impairment and MCI, both conditions that might precede AD development. The reason might be related to inappropriate usage of defense factors, such as pro-inflammatory cytokines, as a result of normally adequate levels of CoA [257,272]. Nonetheless, CoA has high turnover rates in several tissues [253], and its cytosolic and plasma levels might differ from pantothenic acid dietary intake, i.e., a high oral intake of vitamin B5 does not necessarily correspond to higher CoA cellular levels, also determined by the feedback inhibition regulating CoA production [273]. Additionally, it seems more plausible that insufficient CoA levels, essential for acetyl-CoA and, hence, acetylcholine synthesis, might influence cholinergic neurons and in such a way be associated with neurological diseases, including AD [252,274].

In conclusion, a lack of CoA seems to be related to neurodegenerative disorders, due to its antioxidative and anti-inflammatory properties, but its clinical effect has never been demonstrated.

## 8. Vitamin B6 (Pyridoxine)

Vitamin B6 has six different forms in mammals, including pyridoxamine, pyridoxal, and pyridoxine. These three compounds are converted into pyridoxal 5′-phosphate (PLP), which is the biologically active derivative of vitamin B6. PLP is a co-factor for more than 140 enzymes and accounts for almost 4% of humans’ enzymatic activities [275,276]. It is involved in amino acid, glucose, and lipid metabolism, as well as heme and hormone synthesis. In the brain, it is essential for sphingolipid formation for myelin and several neurotransmitters, such as serotonin, norepinephrine, dopamine, and GABA [201,275]. PLP is mostly known for its involvement in the one-carbon metabolism and its link to homocysteine regulation [170,195,276]. Further, B6 is also interacting with the immune systems, through NF-kB downregulation, CRP levels, and peripheric lymphocyte proliferation [195,277,278]. It has antioxidative activities, acting both as a scavenger of reactive oxygen species and as a coenzyme for cysteine, a precursor of glutathione, and it protects blood vessels from AGE accumulation [195,201,276].

Vitamin B6 is mostly available from natural sources, and a varied diet is a sufficient condition for its adequate intake. It is present in meat, fish, nuts, potatoes, bananas, and fortified cereals [275,279].

A severe B6 deficiency is a rare medical condition, leading to microcytic anemia due to an impaired hemoglobin synthesis [201,275]. General symptoms include irritability, depression, migraines, neuropathy, and sleep disorders, and are likely associated with impaired synthesis of neurotransmitters and hormones, especially GABA and melatonin [170,201]. In infants, an inborn error affecting PLP production pathways, might generate acute B6 insufficiency and provoke neonatal epileptic encephalopathies [275]. Marginal and subclinical B6 deficiency is rather common in the general population, and it has been associated with higher cardiovascular risk and chronic inflammatory diseases, including rheumatoid arthritis, chronic bowel disease, diabetes, psychiatric disorders, and cognitive decline [170,277,280]. Several subgroups are at risk of suboptimal B6 levels, independent of their dietary intake, due to increased requests, reduced absorption, or metabolic interactions. Indeed, an inadequate status is frequent in the elderly, as shown by extensive population studies in Europe, UK, and the US, even though body changes more than age itself seem to be determinant, and in alcoholics and women taking low-dosage of oral contraceptives [170,276,281,282].

Vitamin B6 and PLP are deeply associated with normal brain function. Beyond the synthesis of neurotransmitters, B6 compounds are involved in protecting neurons and ECS from oxidative damages and atherosclerosis, maintaining the BBB integrity, regulating inflammatory processes, and regulating homocysteine synthesis [282,283,284,285,286].

Several neurological conditions involve B6 compounds to various extents. Infants and children might develop the so-called pyridoxine-responsive epilepsies, originating mostly from rare recessive mutations in genes codifying for B6 metabolism’s enzymes [287,288]. In animal models with induced Streptococcus pneumoniae-related meningitis, B6 pre-treatment might reduce hippocampal neuronal loss [289,290]. Low levels of B6 seem to be associated with a higher risk of developing PD; it has been argued that PLP is a coenzyme of dopa-decarboxylase, which converts levodopa to dopamine [284,291,292].

The relationship between vitamin B6 and cardiovascular disease (CVD), including significant stroke and small vessel disease, both possibly inducing vascular dementia, depends on various processes in which B6 actively participates. These are inflammation, oxidative metabolism, glycation, and the B6 capacity to lower homocysteine. The latter is a well-known risk factor for endothelial damage and atherosclerosis, and seems to be involved in SVD and subcortical vascular dementia through endothelial damage [293,294]. Indeed, hyperhomocysteinemia in mice, resulting from a diet poor in B vitamins (namely, B6, folate, and B12) led to a vascular dementia model, characterized by microhemorrhages and neuroinflammation [280], and to severe neuronal degeneration, vascular dysfunction, BBB leakage, and memory loss [295]. In addition to lowering homocysteine [296,297,298], B6 might have an independent role in CVD through antioxidative and anti-inflammatory proprieties.

Vitamin B6 protects cerebral endothelial cells from oxidative damage and prevents atherosclerosis and BBB dysfunction. B6 avoids EC apoptosis induced by homocysteine in animal models and reduces endothelial progenitor cells’ apoptosis in stroke patients, likely through inhibition of caspase activity (which in the ischemic condition is enhanced by higher homocysteine levels) [299,300]. Pyridoxamine, pyridoxine, and pyridoxal phosphate reduce superoxide anion and lipid peroxidation, generated by hydrogen peroxide even if it does not prevent the direct cellular damage of these reactive compounds [301]. Pyridoxine interacts with LDL-induced endothelial dysfunction, restoring eNOS activity, NO levels, and cGMP [302,303]. One study reported that subjects receiving supplementation of several vitamins, including B6, had less oxidative stress in the brain measured by the established neural biomarkers [304]. Furthermore, all B6 compounds, particularly pyridoxamine, are active in the degradation of AGE products, reducing the risk of atherosclerosis and the production of free radicals, and perhaps, positively influencing Aβ deposition in AD [193,195,304,305,306].

In the context of inflammation, PLP levels are inversely related to inflammatory markers in several inflammation-related pathologies, such as rheumatoid arthritis and CVD [277]. In particular, low PLP has been related to high CRP in the general population, and high CRP, in turn, is a known risk factor of CVD [285,307,308]. Several studies reviewed by Lotto et al. [277] have found low B6 status in patients with coronary artery disease, myocardial infarction, and stroke, strengthening the hypothesis of an inflammatory link between atherosclerosis-based pathologies and B6. Furthermore, higher intakes of this vitamin have shown protective effects against inflammation. This is not surprising since PLP (B6 active form) influences the cytokine release and the lymphocyte proliferation [278,307,309].

A recent human study showed that in acute ischemic stroke, patients receiving supplementation of B vitamins, including B6, within 12 h after the beginning event, had decreased lipid peroxidation and inflammation, measured as CRP levels, independent from homocysteine levels [283].

Several authors have investigated the association between CVD and low B6 in humans. In a case-control study, low B6 levels were independently associated with the risk of stroke or transient ischemic attacks [310]. In a cohort of patients with silent brain impairments and cognitive decline, brain atrophy was significantly related to low PLP levels, while homocysteine had no significant relationship [311]. Further, in AD patients, white matter hyperintensities (WMH), markers of SVD, were associated with B6 deficiencies, independently of homocysteine concentrations [312,313]. Miller et al. [314] also found reduced levels of PLP in AD patients as opposed to healthy controls. Alternatively, in contrast with these results, Malaguarnera et al. [315] did not find any difference in B6 levels between VaD patients, AD, and healthy controls, but samples were small, and the mean levels of the three groups were not reported [315]. Another study by Nelson et al. [316] did not find any significant relationship between dietary B6 intake and AD risk, but plasma biomarkers were not investigated [285]. Hence, despite some negative findings, a relationship between B6 and dementia progression cannot be excluded.

Several clinical trials tried to assess the benefits of supplementation of B6, B12, and folate on the risk of major vascular diseases, including stroke, coronary artery disease, myocardial infarction, and neurodegenerative conditions, such as AD and VaD, due to SVD.

In the VISP study, Toole et al. [317] enrolled 3680 subjects from Canada, the US, and Scotland with previous non-disabling stroke. Participants received either a low or a high dose of vitamins B6, B12, and folic acid supplementation in a randomized, double-blind procedure. After two years of follow up, the trial was stopped for ineffectiveness: Vitamins, despite the lowering effect on homocysteine, did not reduce the rate of recurrent stroke, coronary artery disease, and death following vascular events [317]. However, several factors might have affected these results, including the deficient baseline homocysteine level, the short follow up time, and the small sample size for the conditions of study [318]. Another multi-centric randomized, double-blind placebo-controlled trial was carried out by the VITATOPS group [319]. Here, 8164 participants with ischemic stroke, TIA, or intracerebral hemorrhage, and from four different continents, were included in the final analysis and had randomly received either placebo or a vitamin supplementation of B6, B12, and folic acid. The primary outcomes were a non-fatal stroke, non-fatal myocardial infarction, and death of vascular origin. There was no significative difference in the two groups after a mean follow up of 3–4 years. However, the authors observed that in a subgroup analysis of patients with symptomatic SVD, the group who received the vitamins seemed to have better outcomes. According to this result, Cavalieri et al. [320] analyzed the effect of the same vitamin formulation versus placebo on MRI changes in patients with cerebral SVD. They did not find any difference in terms of lacunar infarct and WMH in the general group; yet, in a specific subgroup with severe SVD at baseline, the supplementation slowed the progression of WMH [320]. According to this, two other studies on healthy elderly showed that supplementation of B6 and B12 [321] or higher intake of vitamin B6 [322] were related to a higher gray matter volume, in specific brain regions, and also showed better cortical preservation. Another study on healthy and relatively young subjects (mean age around 45 years) investigated possible MRI differences in terms of WMH and brain atrophy, according to the assumption of B6 and folic acid or placebo, over two years. The authors noticed a mild but not significant improvement in MRI parameters in the group with the supplementation [323].

Other studies, involving different vitamin formulation (not including B6), have been conducted and meta-analyzed by the VITATOPS group [319]. Supplementary vitamins do not seem to reduce the risk of CVD in the general population with major vascular events. Nevertheless, particular subgroups, e.g., with severe cerebral SVD, might benefit from vitamin intake and, in the healthy elderly, an adequate vitamin level might help preserve brain integrity.

The association between cognitive function and B6 has been well investigated, with discordant results.

Low PLP levels have been associated with cognitive decline over two years in a Boston-Puerto Rican population, with a stronger link in the older participants [324]. Higher intakes of B6 showed a positive correlation with memory, psychomotor abilities, and verbal fluency [325,326,327]. Conversely, one study on healthy elderly subjects, despite an association between low B6 levels and poor baseline cognition, reported a loss of correlation after adjustment for covariates and other vitamins. Jannusch et al. [322] did not find any link between B6 and cognition in healthy subjects with normal to high levels. Hence, they suggested the existence of a “ceiling effect” [322]: When normal-high vitamin B6 levels have been reached, the effect on cognition after a supplementation might vanish.

Supplementation studies with cognitive function as primary outcome yielded similarly inconsistent results. In patients with mild-moderate AD, a high dosage of B6–B9 and B12, as opposed to a placebo, did not improve or slow down cognitive decline [328,329], despite a definite lowering of homocysteine levels. Similarly, patients with past ischemic vascular events who received B6 supplements did not show any change in cognitive function over 12 months [330]. A recent metanalysis, from 2017 [331], reported the lack of evidence on the therapeutic effects of vitamin supplementation on cognitive function in subjects with cognitive impairment due to AD or other dementias [331]. In healthy subjects, a significant effect on cognition was not reported either, despite a slight improvement in memory, and particularly of long-term memory [332,333,334].

More studies focusing on single vitamins, specific subgroup analysis, and longer follow-ups are still required to prove B6 involvement and potential benefits on cognitive decline. To the best of our knowledge, no study has yet reported breakthrough results, but, in light of current findings, there is a strong need for more specific and dedicated investigation [335].

## 9. Vitamin B7 (Biotin)

Vitamin B7, also called biotin or vitamin H (hair), is widely known for its role as co-factor for four carboxylases: acetyl-CoA, pyruvate carboxylase, beta-methylcrotonyl-CoA carboxylase, and propionyl-CoA carboxylase, essential for the synthesis of fatty acid, amino acids catabolism, the citric acid cycle, energy production, and gluconeogenesis [336,337,338].

Recently, the role of biotin has been extended to epigenetic modulation through histones biotinylation, gene expression, and cellular signaling [338,339,340]. In particular, it has been found that biotin regulates the synthesis of enzymes involved in the glucose homeostasis, such as glucokinase and phosphoenolpyruvate carboxykinase, and stimulates soluble guanylate cyclase (sGC), boosting cGMP concentrations [201,339,341].

Biotin is also involved in inflammatory processes and is a critical factor for NF-κB expression in lymphoid cells [336]. Further, it seems to have a role as an antioxidant, likely interacting with PARP1, reducing pro-inflammatory cytokines and apoptosis, as shown in hippocampal neurons receiving γ-irradiation [342,343].

Biotin levels depend on dietary intake and partially on gut flora production. Various foods are rich in this vitamin, such as liver, egg yolks, soybeans, fish, and leafy vegetables [201,337]. Severe biotin deficiency is uncommon and is usually related to malnutrition or a lack of biotinidase, involved in biotin recycling [338]. Clinical symptoms include alopecia, dermatitis, ketolactic acidosis, skin infection, ataxia, hypotonia, sensory loss, hallucination, and lethargy [337,338,344]. Indeed, neuronal cells and skin cells seem more sensitive to biotin deficiency than other cells, like fibroblasts [345]. Low biotin levels might be found in individuals with type II diabetes or poor glucose-regulation and pregnancy, perhaps due to constant increased metabolic requests [201,337]. Interestingly, unlike other B vitamins, B7 levels in the elderly seem to be in range or even higher as opposed to youngsters [346,347]; hence, more investigations are needed.

Biotin concentrations in the CNS are higher than in the plasma and are located in specific brain regions, mostly the diencephalic area and cerebellar motor nuclei [348]. A saturable transport system handles the transportation of B7 through the BBB, depending on a free carboxylic acid group, which, if necessary, can enhance biotin concentration in the cerebrospinal fluid to levels of 50%–250% more than the plasma [348,349,350]. Indeed, moderate B7 deficiency does not typically lead to neurological symptoms and is related to healthy brain carboxylase function [338].

The most common severe conditions related to biotin lack are the genetically determinant defects in biotin metabolism or transportation enzymes, e.g., biotin-responsive basal ganglia disease. These disorders are rare, mostly affecting children and adolescents, and could be potentially fatal if not adequately replaced with a high dosage of biotin [338,351].

In addition to genetic alterations, biotin might play a role in some chronic diseases and have neuroprotective functions. Indeed, in the 1980s, biotin was reported to improve diabetic neuropathy, as well as uremic neurological complications in a small number of subjects [352,353]. More recently, a study showed that biotin administration in multiple sclerosis patients slows the disability progression and ameliorates the global clinical impression compared with placebo [336]. No benefit was reported in a cohort of older patients with different patterns of progressive MS [354], and more investigations are certainly required.

Two mechanisms might be behind the biotin effect observed in MS. Through the potentiation of the carboxylases activity, biotin might enhance fatty acid synthesis and, thus, increase re-myelination and energy production in neurons, protecting against myelin loss and hypoxia-driven degeneration [354,355]. By a direct stimulation of the sGC, biotin might increase the brain production of cGMP, which is involved in survival, plasticity, and protective pathways [341].

To the best of our knowledge, biotin might play a role in chronic neurodegenerative disorders, such as AD, where patients have low cerebrospinal fluid cGMP levels, as well as in vascular dementia associated with SVD due to cGMP having an anti-inflammatory activity on the cerebral microvasculature, and in stroke due to cGMP anti-atherosclerotic proprieties [341,356,357]. Interestingly, one study on stroke-prone rats reported a decreased systolic blood pressure and a lower stroke incidence (0%) in the animals that received a saline solution with biotin, as opposed to the group that received the solution without biotin (20% of stroke incidence) [344]. Furthermore, B7 might play a role in stroke induction, through the regulation of NF-kB expression, and neuronal apoptosis [338,358].

Biotin might be involved and improve several acute and chronic neurological conditions, but the interest in its neuroprotective function is recent and more studies investigating possible pathways and mechanisms of action in vitro, as well as in vivo, are needed to clarify its role in the brain [359].

## 10. Vitamin B9 (Folic Acid) and Vitamin B12: Separate or Coexistent Realities?

Folic acid derives its name from the Latin word, folium, meaning leaf, and the active form of vitamin B9 is folate known as levomefolic acid or 5-methyltetrahydrofolate (5-MTHF). Dietary folic acid predominantly exists as polyglutamate, which have to be hydrolyzed to mono-glutamates in order to be available [182,360]. Folic acid is hydrolyzed in the gut, and mono-glutamylated folates are absorbed in the duodenum and the first part of the jejunum by a high-affinity receptor, (PCFT1) [361,362]. Folic acid exerts its action mainly through its participation in the so-called histidine cycle (deamination of histidine, production of urocanic acid, and generation of fomiminoglutamate, fundamental for all the glutamic acid cycles), serine and glycine cycle, thymidylate cycle, and purine cycle. A quantity of 5–10 methylene tetrahydrofolate provides a hydroxymethyl group to glycine residues in order to produce serine, the principal donor of the one-carbon unit. The methylation process will be further explained in the text below. Folate is involved in the thymidylate synthase, which is fundamental in cell replication [363]. Finally, tetrahydrofolate derivatives are employed in two reaction steps of the de novo synthesis of purine.

There are different conditions which cause folate depletion; some are physiological, such as aging or pregnancy, while some are para-physiological, such as the recovery from wounds or systemic illness [364,365,366,367]. Dietary low intake, alcoholism, malabsorption, diffuse inflammatory disease of the small intestine, Crohn’s disease, coeliac disease, chronic liver disease, kidney failure, and medications (i.e., phenytoin, carbamazepine, metformin, methotrexate, and sulfasalazine) reduce the activity of pteroyl polyglutamate, a specific hydrolase required for folate absorption, and thereby leading to folate deficiency [368,369,370]. Some specific hematological conditions (occurring as an isolated deficit of homocysteine methyltransferase or a combined B12-homocysteine methyltransferase defect) lead to a clinical situation defined as methyl trap of tetrahydrofolate (THF). This active metabolic form of folic acid is converted to act as a donor of methyl-THF, and acid folic cannot be employed differently [368].

Curiously enough, even vitamin B12, also called cobalamin, is not produced by human beings. Approximately 20 human genes are known to be involved in the absorption, transport, and employment of vitamin B12 acquired from the diet [371]. This aspect gives the reason for the two fundamental cofactor roles of vitamin B12. It is required for the correct functions of the cytosolic methionine synthase and the mitochondrial methyl malonyl-CoA mutase. In particular, cobalamin helps the catalysis process, fundamental for the conversion of L-methyl malonyl-CoA to succinyl-CoA [371]. This reaction is fundamental in the catabolism of the side chain of cholesterol, of the catabolism of the branched-chain amino acids and odd-chain fatty acids. The cytosolic methionine synthase employs the methylated cobalamin for the remethylation of homocysteine to methionine, which we discuss later [372].

A congenital deficit of vitamin B12 is associated with destructive alteration of brain functions [373,374], although different causes precipitate vitamin B12 levels (the most normal aging process, per se) [360]. Some authors converge the aging process with a consequent reduction of intrinsic factor, primary or secondary to atrophic gastritis or hypochlorhydria [373,374]. Other clinical conditions of cobalamin defects are: Genetic deficiency of transcobalamin II, an inadequate intake (vegans, starvation, alcohol addiction, sleeve gastrectomy), malabsorption (inadequate production of intrinsic factors, terminal ileum inflammation, Crohn’s disease, small bowel syndrome, celiac disease), and drugs (pump proton inhibitors, metformin, antiepileptic drugs, chemotherapies, etc.) [369,375]. Vitamin B12 is used by the body in two forms, either as methylcobalamin or five deoxyadenosine cobalamin. The enzyme methionine synthase needs methylcobalamin as a cofactor. This enzyme is generally involved in the conversion of the amino acid homocysteine into methionine, while methionine, in turn, is required for DNA methylation [371]. The cofactor 5 deoxyadenosyl cobalamin is needed by the enzyme that converts l methyl malonyl CoA to succinyl CoA. This stage leads to an extraction of energy from proteins and fats. In addition, succinyl CoA is necessary for the production of hemoglobin, which is the vehicle for oxygen in red blood cells [376]. The loss of vitamin B12 leads to an impairment of Kreb’s cycle, with a reduction of efficacy of the conversion of succinate to fumarate, malate, and the end product of the cycle, rendered less efficacious in energy production [377,378]. Due to the lack of vitamin B12, there is an impairment in gluconeogenesis [379].

Folate and B12 are intimately bound in the methylation reactions, which we discuss in the following section. It is an old axiomatic teaching that treating a B12 deficient patient with folate or conversely a folate-deficient patient with B12 may exacerbate the neurologic consequences of either deficiency; therefore, cyanocobalamin deficiencies should be excluded before folate supplementation is commenced or, if necessary, it should be appropriate to supplement folate and vitamin B12 together [360,380,381,382,383,384].

This axiom is not fully understood; nevertheless, it has been recently rewritten by the NIH [385]. The recommendation implies that supplementation of large amounts of folic acid can mask the damaging effects of vitamin B12 deficiency and, therefore, NIH recommends that folic acid intake from fortified food and supplements should not exceed 1000 μg daily in healthy adults. Other recent data suggested that although an estimated 9–12% of older people in the UK suffering from folate deficiency, due to their everyday dietary intake, there is an excess of folic acid intake in young people, due to promotional and commercial purposes, even if there is “insufficient evidence that folic fortification could promote cancer” [170]. No precise rule has emerged, although as concluded by Porter et al., “those contemplating public health issues worldwide deed to consider a balanced approach and should endeavor to achieve the optimal status of all relevant B-vitamins, throughout all stages of life” [170]. The other emerging problem is new diet habits, i.e., vegetarians, vegans, and patients with an inferior diet (polished rice and nothing more) [386]. In the USA, good clinical practice, as stated by the Institute of Medicine and the Harvard TH Chan School of Public Health [387], recommends a general intake of 400 micrograms per day of folate, suggesting that people who regularly drink alcohol should intake at least 600 micrograms per day. Food sources of folate are fruits and vegetables, whole grains, beans, cereals, and fortified grain products. The recommended intake of vitamin B12 is 2.4 micrograms per day. It is found in fish, poultry, meat, eggs, dairy, fortified cereals, and enriched soy or rice milk [360].

Although many studies have been dedicated to the implementation of folic acid, vitamin B12, or both, in vascular condition, AD or advanced dementia patients, or the old population, de facto results have never been reproducible, and the debate thus remains open [388,389,390,391,392,393,394,395,396,397,398,399,400,401,402,403,404,405].

In conclusion, although the complex metabolic process in which vitamin B12 and folate are bound is well known, there is a lack of knowledge of implementation and time onset of supplementation. It appears as if more evidence would result from more advanced study, but strict lab measures, exclusion criteria, and frequent observation should be ethically recommended rules in clinical practice [360,406].

## 11. Homocysteine: The Sharing Process of Vitamin B12 and Folate.

Homocysteine (Hcy) is related to the production of 5,10-methylenetetrahydrofolate, a fundamental step for the synthesis of thymidylate, purines and methionine, employing vitamin B12 and folate as cofactors [407,408,409,410]. The S-adenosyl-methionine (SAM) to S-adenosyl-L-homocysteine (SAH) ratio defines the methylation potential of a cell [411]. If homocysteine is allowed to accumulate in normal conditions, it will be rapidly metabolized to SAH [412]. Whenever there is a methionine deficit, Hcy can be re-methylated to form methionine, by the employment of N5, N10-methylenetetrahydrofolate [412]. If there is an adequate amount of methionine, Hcy is employed for the production of cysteine, mediated by cystathionine–beta-synthase, with pyridoxine and folate as a cofactor [182]. Therefore, the accumulation of Hcy is dangerous when it occurs in the absence of folate and vitamin B12 as a cofactor. The causative factors of accumulation of Hcy in healthy adult life can be diverse, due to various genetic defects, or to the defects of vitamin B12 and folate [182]. A physiological increase of Hcy occurs in the brain (and CSF) and the plasma, within the aging process [296]. Evidence has showed that the adult life hyperhomocysteine condition is associated with cardiovascular and cerebrovascular diseases [413,414]. It has been reported that hyperhomocysteine (HHcy) promotes cerebro/cardiovascular atherosclerosis and instable/ruptured atherothrombotic plaques, upregulating the expression of matrix metalloproteinases-9 (MMP-9) expression [415,416].

Homocysteine is a common final pathway of the lack of folate and vitamin B12. Its role has been widely discussed elsewhere [296]; here, we report the significant and more innovative lines of study of homocysteine in brain vascular alteration, underlying small vessel diseases.

Undoubtedly, the methylation reactions are necessary for the brain, SAM being the sole donor in numerous methylation reactions involving proteins, phospholipids, and biogenic amines [375], and for packaging of many phospholipids [417]. Most of the polyunsaturated phosphatidylcholines (PC) in mammals are synthesized by phosphatidylethanolamine N-methyltransferase (PEMT) [418,419,420], which methylates phosphatidylethanolamine to generate phosphatidylcholine using SAM as the methyl donor. The leading site of the PEMT-catalyzed synthesis of PC-DHA is liver, whereas the PEMT activity in the brain is relatively low [421], and the polyunsaturated species of PC [422] that are synthesized there are adequate for local needs only. PEMT activity in the brain is higher during the perinatal period and declines in adulthood [421].

The PEMT reaction consumes three molecules of SAM for every PC molecule produced and generates three molecules of S-adenosyl-L-homocysteine (SAH), which act as an inhibitor of PEMT [423]. Some of the SAH-derived hepatic homocysteine enters the circulation, determining a link between Hcy and PEMT [424]. Some studies of AD patients related higher plasma concentration of Hcy to reduced levels of erythrocyte phosphatidylcholine-docosahexaenoic acid [425]. The role of phosphatidylcholine has been debated in vascular dementia, and citicoline has demonstrated neuroprotective effects in acute stroke and has been shown to improve cognition in patients with chronic cerebrovascular disease and some patients with Alzheimer’s disease. Citicoline has prevented cognitive decline after stroke, exhibiting a neuroprotective effect [426,427,428,429,430].

It has been proven that Hcy could be linked to neurodegeneration; Hcy, together with high levels of glycine in the brain, is an agonist of the endogenous NMDA receptors [431], influencing calcium influx [432], and exerting a direct activation of the group I metabotropic glutamate receptors [433]. Hyperhomocysteine (HHcy) upregulates Presenilin 1, which promotes APP synthesis [434]. The protein phosphatase methyltransferase 1, whose methylation is SAM-dependent, regulates the activity of the protein phosphatase methyltransferase 2A, which acts as a dephosphorylating system for tau protein [435]. Hence, the reduced methylation capacity increases the hyperphosphorylation of tau protein, determining microtubule disaggregation, and the deposition of the neurofibrillary tangles. Moreover, Hcy potentiates the toxicity of Abeta 42 deposition [436], in particular, increasing its deleterious effects on the smooth vascular cells in the brain [437].

Hcy acts as a pro-inflammatory and pro-oxidative factor. The SAM-to SAH ratio is the expression of the methylation potential of a cell; as a consequence, “HHcy tends to decrease the methylation potential” [412]. Therefore, Hcy can induce a global DNA hypomethylation and suppress the transcription of cyclin A in endothelial cells. Apoptosis of human endothelial cells after growth factor deprivation is associated with rapid and dramatic up-regulation of cyclin A–associated cyclin-dependent kinase 2 activity. Simultaneously, Hcy leads to up-regulation of other genes, causing an increase of p66shc expression in endothelial cells, inducing oxidant stress [412]. HHcy leads to an induction of mRNA of C-reactive protein (CRP), augmenting the NR1 subunit of NMDA receptor expression. Therefore, Hcy can promote a pro-inflammatory response in vascular smooth muscle cells of small brain arteries by stimulating CRP production, usually enhanced by a combined NMDA-ROS-erk1/2/p38-nfKBeta signal pathway [438].

Recently, a well-conducted study [439] demonstrated that cultured cell incubation with Hcy determined cell death at 80 microM Hcy exposure after five days; impressively, cell exposure to Hcy at lower concentrations for five days increased reactive oxygen species (ROS) production 4.4-fold. HHcy induced, in the beginning, endothelial cells to produce nitric oxide (NO) and to yield S-nitrose-Hcy, which acts as a protector of endothelium, although the chronic exposure to Hcy induces a final diminishment of NO [440]. HHcy acts in a multistep process against the endothelium. Its accumulation leads to a disruption of the cell’s integrity and, then, the HHcy-dependent reduced NO bioavailability induces an altered endothelium relaxation and an urgent inflammatory response of muscle cell arteries, testified by an increase of C-reactive proteins and cysteinyl-leukotrienes and of HMG-CoA reductase [441]. Finally, HHcy tends to reduce the efficiency of the cystathionine-beta-synthase, which induces altered redox homeostasis, with macroscopic alteration of the oxidative repairing process [442,443]. An accelerated lipid peroxidation sequela is the main result, with fatal outcomes for neuronal cells, astrocytes, and neurovascular coupling [444,445,446,447,448,449].

Recently, HHcy by itself (without folate and B12 deficiencies) has been associated with secondary septic status and conditioned poor outcome [450]. It has been argued that it leads to a direct increase of the macrophage response, with an induced release of ROS, and an altered oxidative repair [450]. More recently, an induction of B-lymphocyte has been demonstrated by a HHcy, with a macroscopic stimulation of the activity of the pyruvate kinase muscle isozyme 2, which seems to be involved in the acceleration of the atherosclerosis process [451,452] and probably the alteration of the transcriptional repression of fibroblast growth factor 2 [428]. As explained [296], the promoted HHcy activity on NMDA receptors has been demonstrated not only on neurons but also on neutrophils and macrophages. In this way, HHcy upregulates the nuclear factor-kappa B, one of “the master regulators of the expression of inflammatory genes” [453,454].

Even all these positive data, clinical trials, and studies failed to demonstrate conclusive results, either considering HHcy as a deterrence target, or preventing HHcy through supplementation in patients or a healthy population with vitamin B12, folate, or both. Many criticisms may be made of the trials that have been implemented [455,456,457,458].

## 12. Conclusions

The conclusions of the present review can be stated as follows: Vitamins of the B group are tightly related to gene control for endothelium protection and act as antioxidants. They play a co-enzymatic role in a number of ways: conversion of pyruvate to acetyl-CoA; conversion of alpha-ketoglutarate to succinate, in the Kreb’s cycle; catalysis by transketolase in the pentose monophosphate shunt, superoxide dismutase (SOD) and catalase; and, in the one-carbon metabolism cycle, to reduce 5,10-methylenetetrahydrofolate (5,10-MTHF) to 5-methyl THF, which, in turn, provides the methyl group for homocysteine re-methylation to methionine pantothenate kinase-2, acetyl-CoA, pyruvate carboxylase, beta-methylcrotonyl-CoA carboxylase, and propionyl-CoA carboxylase). Often, their role is the most critical in many different biochemical reactions inside the brain, interacting with many other constituents, such as participating in the synthesis of polyunsaturated phosphatidylcholine, through the SAM methyl donation (Figure 1). They have anti-inflammatory properties and play a protective role against neurodegenerative mechanisms, such as by modifying the glutamate currents and reducing calcium currents, as well as showing important antioxidant properties. It has even been argued that they play a direct neuropeptide synthesis promoter role.

However, there remain many coincidences and a significant number of unanswered questions [459,460,461]. Laboratory data are far from providing a clinical resolution of these mysteries, such as in the cases of AD or small vessel disease dementia. However, “one coincidence is just a coincidence, two coincidences are a clue, three coincidences are a proof”; at the moment, all the laboratory analyses are directed without any hesitation towards the data mentioned above. The questions are:Should we apply them to clinical practice?When should we employ them? Earlier is better, but when is “early” not intelligent or too wasteful?How can we manage clinical trials to be simultaneously efficacious and objective?Which are the markers of the evolution from healthy aging towards pathology? In this case, a question that remains open concerns identifying when the small vessel alteration of white matter becomes dementia.

Further studies should take into account all these questions, with well-designed and globally homogeneous trials.

## Figures and Tables

**Figure 1 ijms-20-05797-f001:**
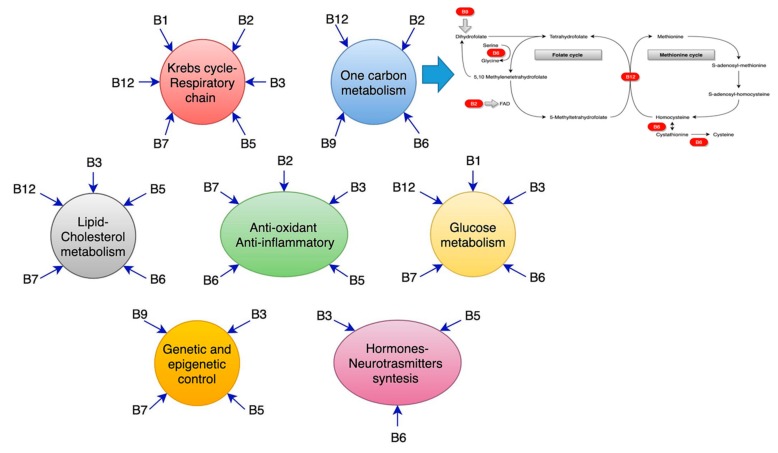
A synopsis of the different biochemical pathways supported by vitamins B.

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
