# Peer review of "B Vitamins and Fatty Acids: What Do They Share with Small Vessel Disease-Related Dementia?"

_ijms, 2019, doi:10.3390/ijms20225797_

Round 1

Reviewer 1 Report

The paper from Moretti R. and Peinkhofer C. is an extensive review that addresses in a complete and in-depth way the role of B vitamins on the onset, pathogenesis but also prevention of vascular dementia and in particular small vessel disease-related dementia. The work is well written and clear in its logic of development. The authors have done a comprehensive analysis of the available literature, ranging from basic science to clinical trials. The language is adequate, even if some typing and punctuation errors should be eliminated. I believe that already in the present form the work can be published, without the need for particular revisions. A doubt concerns the first sentence of the conclusions in which it is stated that "vitamins of the B group ... .play and enzymatic and often co-enzymatic role in the most critical etc". Could the authors make the meaning of this sentence more clear, please? I have some doubt about the enzymatic role of B group vitamins.

Author Response

Thank you for your comments and your encouragement. 

We have rewritten the first sentence of the conclusions in which it is stated that "vitamins of the B group ... .play and enzymatic and often co-enzymatic role in the most critical etc". 

Of course, it was a mistake: we have rewritten it as follows:

"All these words conducted invariably to one direction; vitamins of the B group are tightly related to gene control for endothelium protection, act as antioxidants, play a co-enzymatic role (conversion of pyruvate to acetyl-CoA, the conversion of alpha-ketoglutarate to succinate, in the Krebs cycle, and the catalysis by transketolase in the pentose monophosphate shunt , superoxide dismutase (SOD) and catalase, one-carbon metabolism cycle, to reduce 5,10-methylenetetrahydrofolate (5,10-MTHF) to 5- methyl THF which, in turn, provides the methyl group for homocysteine re-methylation to methionine pantothenate kinase-2, acetyl-CoA, pyruvate carboxylase, beta-methylcrotonyl-CoA carboxylase and propionyl- CoA carboxylase ); often, their role is the most critical in many different biochemical reactions inside the brain, interacting with many other constituents, like participating in the synthesis of polyunsaturated phosphatidylcholine, through the SAM methyl donation. They have anti-inflammatory properties and act as a protective role against neurodegenerative mechanisms, such as modifying the glutamate currents and reducing calcium currents, as well as showing important antioxidant properties. For different of them even a direct neuropeptide synthesis promoter role has been argued. " 

THANK YOU AGAIN

Reviewer 2 Report

While the review is of interest to the field, in its current it is very difficult to read.   Some sections are well written but other sections requires major editing of both English language & style.  The general content of the review is logical, but a review should be up to date & focus on recent major findings which have contributed to the field.  While key papers are identified, this review lacks more recent, current studies.  Where appropriate, a relevant figure could be included to compliment the text.

Author Response

While the review is of interest to the field, in its current it is very difficult to read.   Some sections are well written but other sections requires major editing of both English language & style. 

Thank you for your help and advice: we have rewritten many paragraphs and sentences in the text, highlighted in green, as you will see, hoping that the result would be better.

The general content of the review is logical, but a review should be up to date & focus on recent major findings which have contributed to the field.  While key papers are identified, this review lacks more recent, current studies.  Where appropriate, a relevant figure could be included to compliment the text.

Thank you for your advice. We have re-revised the Literature on the topic; as you can see, it contains now 461 references, 156 of them written from 2017-up to 2019. Some of the newest and of the most relevant, have been added and they are highlighted in blue throughout the text.

THANK YOU

Round 2

Reviewer 2 Report

The manuscript is greatly improved by the addition of more recent, relevant references.  However, while I appreciate that some sections have been rewritten to improve the writing style, the overall quality of the writing is still well below what is acceptable for publication.  For example, in the abstract "and any real conclusion has been drawn" makes no sense; and the sentences are short & disjointed.  Also, the authors refer to "Pandora's vase" when they should have stated "Pandora's box".  These are just some of the examples of poor quality English on the first page & the manuscript does not greatly improve throughout the text.  

Author Response

The manuscript is greatly improved by the addition of more recent, relevant references.  However, while I appreciate that some sections have been rewritten to improve the writing style, the overall quality of the writing is still well below what is acceptable for publication.  For example, in the abstract "and any real conclusion has been drawn" makes no sense; and the sentences are short & disjointed.  Also, the authors refer to "Pandora's vase" when they should have stated "Pandora's box".  These are just some of the examples of poor quality English on the first page & the manuscript does not greatly improve throughout the text.  

Dear Sir,

thank you for your appreciation for our work on the more recent reference.

The work has been re-read by us; and then an English native speaker, Mr. Andrew Rosenberg, Ph.D., has corrected it entirely.

You will find the renewed parts of the ms written in italics.

Thank you again

Rita Moretti

Round 3

Reviewer 2 Report

In their response the authors state that 

"The work has been re-read by us; and then an English native speaker, Mr. Andrew Rosenberg, Ph.D., has corrected it entirely.  You will find the renewed parts of the ms written in italics"

While I commend the authors for inviting someone with English as their first language to read the manuscript, it has not improved the English language & style.  I am very disappointed that the authors claim that all renewed parts of the manuscript are in italics.  Their entire manuscript is in italics, suggesting that it has been entirely rewritten.  But this is a blatant lie on the part of the authors & I am shocked and deeply disappointed that they would make such claims.  The majority of the sections in italics have not changed in the slightest from their previously submitted draft.

Round 4

Reviewer 2 Report

English language & style greatly improved.  The manuscript is now acceptable for publication